# On the Road of Discovery with Systemic Exploratory Constellations: Potentials of Online Constellation Exercises about Sustainability Transitions

**Antje Disterheft** [1,*] , **Denis Pijetlovic** [2] **and Georg Müller-Christ** [2]

1   CENSE—Center for Environmental and Sustainability Research, NOVA School of Science and Technology, NOVA University Lisbon, Campus de Caparica, 2829-516 Caparica, Portugal

2   Faculty 7: Business Studies and Economics, University of Bremen, Enrique-Schmidt-Straße, 28359 Bremen, Germany; denis.pijetlovic@uni-bremen.de (D.P.); gmc@uni-bremen.de (G.M.-C.)

*   Correspondence: a.disterheft@fct.unl.pt

**Abstract:** Sustainability transitions are shaped by specific dynamics, dependencies, and influences among the actors and elements that are part of the system. Systemic constellations as a social science research method can offer tangible visualizations of such system dynamics and thereby extract valuable, often hidden knowledge for research. This article builds on two online exploratory system constellation exercises about sustainability transitions, with two major objectives: (i) to introduce and disseminate (exploratory) systemic constellations as a method for (sustainability) research, and (ii) to extract their potential for (online) collaborative and transdisciplinary research, with a focus on sustainability transitions. Our exploratory research design includes participatory action research that took place during the virtual International Sustainability Transitions Conference 2020, Vienna, Austria. Data were analyzed following an interpretative-hermeneutic approach. The main findings consist of visualizations about sustainability transition dynamics between selected actors in Germany and Portugal that are discussed in light of the literature on constellation work and sustainability transitions, triggering new assumptions: (i) a strong sustainability narrative does not (necessarily) lead to action and transformation and (ii) transformation requires integrating narratives beyond weak and strong sustainability. We conclude with a list of potentials of exploratory constellations for sustainability research and online formats that offer novelties such as a constant bird-eye perspective on the system while simultaneously engaging with the system.

**Keywords:** sustainability transitions; action research; (exploratory) systemic constellations; emerging research; narratives; collaborative approaches; virtual formats



## 1. Introduction

Sustainability transition processes are shaped by and depend on multiple factors, with specific dynamics, dependencies, and influences among them. To tackle the complex global problems the world is facing [1–3], sustainability transitions strive for societal transformation and hence require collaborative efforts, on the individual and collective level [4] with multiple challenges regarding politics and governance, civil society and culture, and businesses and industries [5].

Sustainability transitions have become not only an important academic field within Sustainability Science, but overall a field of joint experimentation for science and society, bringing together practitioners, citizens, enterprises, governmental entities, and academics. Engaging in and mobilizing for these collaborative efforts for systemic and structural change is, however, a difficult task, considering the heterogeneity of societies and the existing clashes and divisions among societal groups. Adopting a systemic perspective can be useful to better understand the critical conditions that would enable individual and collective actors to build up and move forward the capacity of the system towards

sustainability goals; but what are the critical conditions that enable individuals and collective actors to build up and mobilize the capacity for systemic and structural change? This was one of the guiding questions of the International Sustainability Transition Conference (IST) 2020, organized and hosted in August 2020 by the Austrian Institute for Technology and Vienna University of Economics and Business, that inspired us to propose an exploratory systemic constellation exercise within one of the dialogue sessions offered during the conference. Systemic constellations can be broadly described as "the physical representation of a system by means of representatives. These are real people who can be set up to represent different aspects of the system under investigation" [6]. Offering a constellation exercise during the IST 2020 would allow for creating a tangible visualization of a systemic perspective on sustainability transitions. Furthermore, such an exercise could be very helpful and powerful to look at the dynamics of a system, learning in more detail about specific elements and unlocking hidden knowledge. Additionally, constellations offer a great potential to dive into deep and meaningful exchanges in a short timeframe, beyond hierarchies and professional backgrounds.

Even though systemic constellations have already been used for about 5 decades (more in-depth information in Section 2), only in more recent years have they entered the academic world and are being used in research [7–13]. Since the pandemic due to the COVID-19 crisis, constellation practitioners have started to adapt their work to virtual formats. This article builds on two constellation exercises about sustainability transitions—a pre-test and the exercise during IST 2020—that were both forced to happen online because of the already mentioned pandemic, and presents therefore a novelty, being one of the first publications about online constellation work in research. We follow two major objectives: (i) to disseminate (exploratory) systemic constellations as a method for (sustainability) research, with a focus on links to sustainability transitions; and (ii) to extract their potential for scientific purposes, with a focus on virtual environments.

First, we provide the theoretical context of exploratory research and exploratory systemic constellations as a research method, linking it to the theoretic fields of sustainability transitions as the framework for the constellation set up (Section 2). Next, we explain our methods (Section 3), followed by our findings of the constellation exercises (Section 4). In Section 5, we discuss our findings and establish links to sustainability transitions research, pointing towards new emergent questions and assumptions. We conclude with a summary of the potentials for (online) constellation work in research and practice.

## 2. Theoretical Context: Systemic Exploratory Constellations and Sustainability Transitions

As soon as one investigates the potential of a matter, such as transformation to sustainability, a space of possibility opens up that relates to a future that can be explored. How can the potential of transformation for sustainability be explored using scientific methods? Exploratory research with constellations is one possibility we would like to present [10,14]. Exploratory research using systemic constellations is a holistic view of systems that allows for multi-contextuality. In this context, multi-contextuality means that several contexts can be observed simultaneously with the help of a system constellation.

### 2.1. What Are Exploratory Constellations?

The method of exploratory constellations is assigned to systemic research and integrates various methodological approaches such as knowledge procedures, interview procedures, participation procedures, and hermeneutics. Exploratory constellations can be used to test suspicions and assumptions about the extent to which existing theory applies to the observation, explanation, and design of systems, which elements characterize the shape of a system, and what non-visible causal structures are located beneath the surface of a system [10] (p. 11). Basically, exploratory constellations are one type of systemic constellations (Figure 1), and, with reference to Carl Rogers [15] (p. 12), are by no means an objectively new or unknown method. Historically, systemic constellations have been used since 1990 [16] (p. 10), [8] (p. 64), first in the context of family therapy (invented by Virginia

Satir) and self-awareness, and later in organizational consulting, management, training, and coaching [17] (Figure 1). What is relatively new, however, is that systemic constellations in the form of exploratory constellations are used in colleges and universities as a research and teaching method to generate data and insights. In principle, constellations allow the spatial representation of a system with the help of representatives. These are real people who can be constellated to represent various aspects of the system under investigation [6]. The method thus offers the possibility to externalize, view, and edit the inner image of a person or group of his and/or her system with the help of representatives [18] (p. 71). This image in space, externalized by means of the representatives, creates a projection surface for the viewer, which makes it possible to gain new perspectives and insights about the system. Another basic principle is that constellations are guided by constellation facilitators and they usually take place in a group of about six to 12 people. In classical constellations, concerns, i.e., a question or problem, are usually dealt with by one person. Representatives are selected for all relevant aspects of the system. The representatives use their perception as a language for the element they represent [8] (p. 65). Exploratory constellations make it possible that (i) a system can be viewed from different perspectives, and (ii) dynamics and development paths of the system become visible.

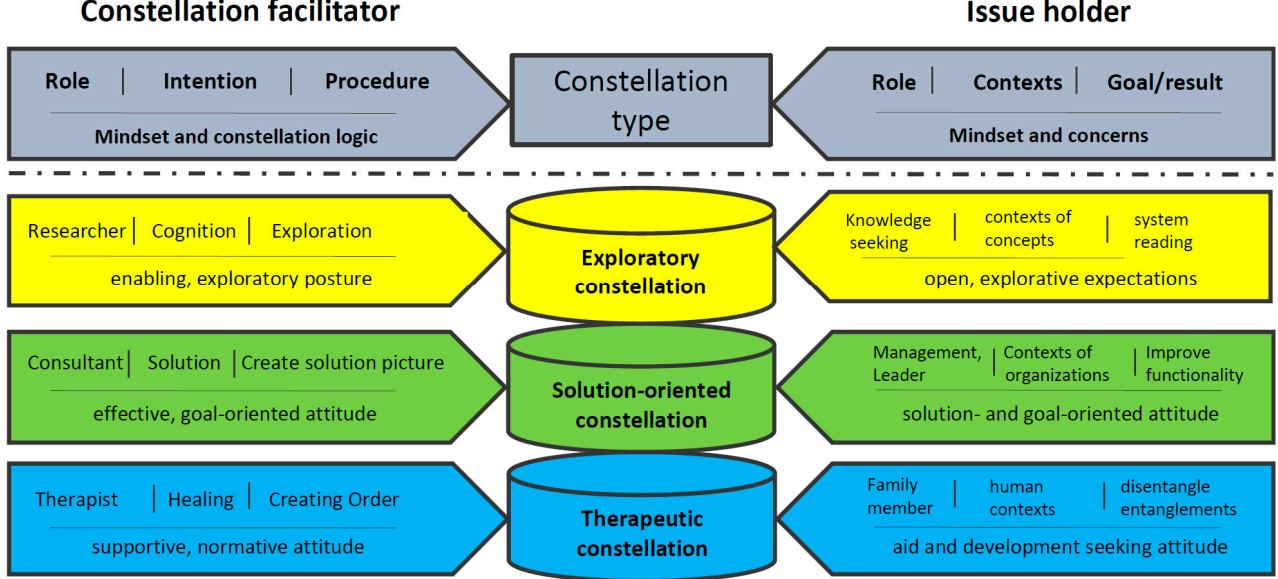

**Figure 1.** Types of system constellations [13] (p. 131). Notes: On the left side, the graphic shows the role of the constellation facilitator, and on the right side the role of the issue holder. Both sides must fit together according to this model, as otherwise the objectives for the constellation are not aligned and can bring the facilitator and issue holder into conflict.

In contrast to other types of constellations, which are hypothesis-driven and seek to change systems, exploratory constellations describe more the developments, dynamics, and driving forces from which a particular system picture results, without seeking to change it through hypotheses and interventions (Figure 1).

Exploratory constellations can be used for different objectives, and their function can ideally be described as explorative knowledge and cognition expansion that can have a positive effect on communication, goal concretization, and goal formation, as well as on decision making and strategy building. Such expansion of knowledge and cognition is primarily sparked by strong visualizations and images emerging from the constellation process and by deepening the existing understanding of the constellated system. The topics to explore with this type of constellation can be theoretical or prototypical concepts and ideas that are set up. Since exploratory constellations are based on the assessment of relevant elements and logical polarities, they force the explication of existing implicit or even unconscious basic assumptions about system states. Here, the constellation facilitation

takes more the role of a researcher, who observes and extracts as much possible data and information from the constellation without intervening, in order to enable a large variety of insights. The underlying intention for this procedure is to confound the issue holder or the observer in a way that a reflection process is triggered and that from the clashes of personal mental models and different perspectives, something new can be created. Thereby, the three following points are to be considered.

### 2.1.1. First Order Contingency

First, exploratory constellations do not represent a comprehensive picture of system states, because their function is to direct perception specifically to one or more specific, delimited sections of reality. Various polarities and elements are deliberately included (and others excluded) and placed in specific constellations with respect to each other. The purpose of this constellation work is not to depict the state of a system as a whole, but rather to focus on certain aspects of interest by dealing with a particular area of investigation, which may be quite abstract. Luhmann [19] describes contingency as something that is neither necessary nor impossible. Accordingly, something as it is, as it was, as it will be, can also be possible in a completely different way. The term thus designates what is given (to be experienced, expected, thought, fantasized) with regard to a possible otherness. The concept designates objects in the horizon of possible variations [19] (p. 152).

### 2.1.2. Second Order Contingency

Secondly, it should be noted that the constellation setting, i.e., the selection and combination of polarities and elements with respect to the exploration is design work. In doing so, certain stakeholders, contexts, and system forces are deliberately considered relevant or neglected and these are then in turn, under certain assumptions and with empirical knowledge, put into interaction and context. However, a constellation setting can always be constructed differently. Assumptions about the relevance of elements, polarities, and contexts for the scope of investigation of an exploratory constellation are more or less suggested by data, theories, or statements of an expert or issue holder, but on the one hand, they require well-founded knowledge, particularly experience knowledge, and on the other hand, they are usually based on subjective assessments. This can be different if the constellation setting for exploratory constellations is created in a collaborative process in preliminary discussions with a group. Then, the basis of the constellation setting shifts more in the direction of a group-supported objectivity in the sense of a shared reality.

### 2.1.3. Explication

Thirdly, such a sense of a shared reality is connected with the fact that every exploratory constellation is based on implicit assumptions about how the system state might look like, how certain elements behave, which dynamics prevail, which developments remain constant and which change. These implicit assumptions refer to comprehensive mental models and maps, which are usually based on personal experience. With the help of exploratory constellations, these implicit mental maps can be made explicit, but also be confounding, allowing new perspectives and points of view to emerge. It is important to note that exploratory constellations do not claim to be truthful and thus do not provide objective knowledge about the systems that have been constellated. Instead, only the hypothetical construction of possible system states, based on present and past knowledge, and coupled with the influence of "controlled perplexity or discomposure" (i.e., when personal worldviews and mental models clash with different perspectives), can create strong and impactful visualizations of the hypothetical system state. These visualizations can lead to the constructive expansion of mental maps, so that new ideas, thoughts, attitudes, and solutions can emerge with respect to the topic under research.

However, exploratory constellations do not only serve to produce knowledge and insights, but also to uncover the limits of knowledge such as uncertainties, gaps, dilemmas, ambiguities, and complexity. Accordingly, exploratory constellations have a transformative

potential. An initially unknown system space can be transformed into a space of possibility and future. For these reasons, they fit very well into sustainability transitions and transformation research.

### 2.2. Sustainability Transitions and Societal Transformation: Moving from Weak to Strong Sustainability Narratives

The main motivation for research on sustainability transitions and transformation is the recognition of the links between environmental problems (such as climate change, loss of biodiversity, and resource depletion, e.g., clean water, oil, forests, and fish stocks), and human actions (such as unsustainable consumption and production patterns, e.g., electricity, heat, buildings, mobility, and agro-food), urging grand societal challenges to keep the earth in a livable and balanced state [5]. Sustainability science scholars agree that these problems cannot be addressed by incremental improvements and technological fixes, but require radical shifts to new kinds of socio-technical and social-ecological systems. Some research communities call such shifts sustainability transitions [20,21] whereas others emphasize them as sustainability transformations [22–25]. While there are etymological differences between the terms transition and transformation and eventually different concerns regarding scales, both research communities apply systems thinking, share similar goals, and have come closer in the recent past [24]. A central aim of transitions research has been to conceptualize and explain how radical changes can occur in the way societal functions are met. Köhler et al. [5] offer a comprehensive review on the research developments in the sustainability transitions field, highlighting its origins in innovation studies and the systemic perspective that underlie the four main theoretical frameworks of transition studies, which are the Multi-Level Perspective (MLP), the Technological Innovation System approach (TIS), Strategic Niche Management (SNM), and Transition Management (TM). Their review synthesizes that in the past decade, "sustainability transition studies have diversified significantly, with new sub-themes emerging such as urban transitions, acceleration, system decline, system re-configuration and interaction between multiple innovations, ethics and justice, the role of users, power relations in governance structures, and transitions involving multiple sectors. These themes point to the field's continued expansion and demonstrate the usefulness of the broad transition framing when reflecting on the dynamics of radical socio-technical change" [5] (p. 21). Loorbach et al. [25] distinguish further between socio-technical approaches, socio-institutional approaches and socio-ecological approaches in sustainability transitions research and understand dynamics of societal transitions as iterative processes of build-up and breakdown over a period of decades. While different foci and analytical lenses can be set, all approaches apply a high variety of different methods in sustainability transitions and transformation studies, e.g., case studies and engaged transdisciplinary action research [25,26]. Diverse dilemmas have been identified; for example, the level of analysis—micro versus macro—the type of knowledge generated—in-depth particularity versus generic insights—and dealing with complexity—reduction versus articulation [5]. Furthermore, scholars draw the attention to leverage points for sustainability transformation [27], as (sustainability) science has so far failed to engage with the root causes of unsustainability. These scholars suggest a research agenda that would embrace (i) reconnecting people to nature, (ii) restructuring institutions, and (iii) rethinking how knowledge is created and used in pursuit of sustainability. Fazey et al. [28] distinguish between first and second-order transformation research and propose ten essentials for the latter that could "help accelerate learning and actions that lead to transformations towards a low-carbon, resilient and sustainable world". Resulting from a 2-year-long individual and collective reflection process that was initiated by a shared frustration about low acceptance of second order science, the authors challenge several assumptions of traditional roles and knowledge productions, e.g., the assumption of researchers being independent observers as "researchers are inevitably embedded within, and not separate from, the systems they seek to observe" (ibid., p. 56). Their essentials for action-oriented research for sustainability transitions and transformations include, e.g.,

"seek to transcend current thinking and approaches", "take a multifaceted approach to change", and "be reflexive" (ibid., p. 60).

While sustainability transitions and transformation studies argue for rather radical shifts, different understandings on sustainability remain [29].

In the sustainability literature, scientists distinguish between weak and strong sustainability [30–34]. A weak sustainability narrative builds on technological progress and technical solutions, for which natural capital can be substituted by other forms of (human-made) capital, focusing on human welfare and economic growth. A strong (and regenerative) sustainability narrative, though, rejects the substitutability debate and continuous economic growth, recognizing the intrinsic value of nature and focusing on societal transformations together with technological solutions that would allow a good life for all, including non-human living beings.

We found the distinction between weak and strong sustainability narratives useful for the exploratory constellation exercise, as it would allow a visual positioning of certain actors in relation to these concepts in a specific system (context). Moving from weak to strong sustainability should serve as a symbolization for a sustainability transition within a specific context, e.g., a country, that would represent parts of the chosen system (see Sections 3.1 and 3.2). Following the leverage points for sustainability transformation by Abson et al. [27] stated above, we wanted to include, at least, entities as representative elements in the systemic constellation exercise that somehow would refer to those three leverage points, such as (i) society (as a central element for reconnecting people to nature), (ii) politics/governance (as a central element for restructuring institutions), and (iii) science (as a central element for rethinking how knowledge is created). We also understand these entities as actors influencing the structural and framing conditions in a system. However, we recognize that we rather opt for an epistemological than an ontological approach in this set up, based on Abson et al. [27] (p. 32), who state that "a system is [...] defined by the subjective interests and pre-analytic assumptions of the researchers, with all the potential problems this entails". Sustainability transformations researchers acknowledge that systems are therefore also "systems of interest" [35] that are, to some extent, shaped by the worldviews and concerns that researchers and other actors involved hold towards the topic of research.

## 3. Methods and Research Design

We adopted an exploratory research design mixed with participatory action research [36], being the method of system constellations part of the action research process. The research process consisted of the following steps. First, we reviewed the literature on the topic for the conference abstract submission. After the abstract's acceptance, we undertook a reflective dialogue process to discuss the constellation set up and how to introduce systemic constellations to the conference audience new to this method. The first set up was then tested in a pre-test on 19 August 2020 with a group of ten constellation practitioners (partially working professionally with constellations) coming from mixed academic and professional backgrounds (i.e., academic researchers, consultants, coaches, engineering and healthcare professionals). The pre-test revealed a too complex set up with too many representative elements for the conference format and was modified accordingly to fit the time limits of 1.5 h. The final constellation exercise included a short feedback and reflection round with the participants. Further reflection rounds were held after the event between the authors and with the pre-test group of constellation practitioners.

### 3.1. Research Design of the Online Constellation

The final exploratory constellation exercise took place on 21 August 2020 during an online dialogue session at the International Sustainability Transition Conference (IST2020), with 14 full participants along the whole session. The conference used the online events platform *Hop In*, which allowed 10 participants to connect via video/audio streaming

and unlimited attendance of participants to follow the video/audio streaming, including instant messaging (chat) for all.

The main aim of the online exploratory constellation was to find out about dynamics in sustainability transition processes; in particular, about certain actors and their needs to move from a weak to strong sustainability narrative where the strong sustainability narrative stands for structural change and transformation (see Section 2.2). In addition to the focus of the content, the method of the constellation was tested for the first time at a scientific conference as an online format. Table 1 summarizes the research design. Based on the findings from the pre-test, the chosen actors to be represented were *Science*, *Society/Citizens*, and *Politics/Governance*, as these were considered to relate to the leverage points in sustainability transformation research (see Section 2.2). In the pre-test, three more actors were included, namely non-governmental and civil society organizations (NGOs/CSOs), enterprises and companies, and the media, but due to time limits they needed to be excluded. The task of the representatives was to position themselves with an icon in the continuum between *Weak Sustainability Narrative* and *Strong Sustainability Narrative* in two different contexts. For this purpose, we prepared a shared Google Slides file, with slides for each context that could be accessed and edited simultaneously by the representatives, and they could move around their respective icon (see Table 2, image in the left column). The process work is based on differentiations, so the representatives intuitively articulate differences on the positions of their icons. The data collection included observation protocols and field notes, screen shots, the session chat, participants' feedback, and notes of joint reflection rounds after the event (see Table 1). Unfortunately, the online constellation was not completely recorded on video by the conference organizers as planned due to a technical problem with the recording function that turned off automatically without notice before the end of the session.

*3.2. Constellation Format, Process and Data Analysis*

After a general introduction on systemic constellations and their potential use in sustainability transitions research, the participants were given specific instructions for those who would like to volunteer as representatives in the exercise and for the remaining audience (Table 2). The chosen constellation format was double-blind, as this format helps to avoid personal mental models being activated during the constellation and disturbing the intuitive process of the representatives. Furthermore, this format sets against the doubt that representatives would enact a specific role as on stage. By not knowing who or what they represent nor the contexts, research in systemic constellations has shown that participants as representatives find it liberating to only focus on their perception and senses, with no need to think or reflect on their roles [10].Therefore, in line with the double-blind format, the three participants who took part in this exercise as representatives neither knew about the context nor which element they represented (Table 2). They only knew that the general topic of the constellation referred to the conference theme sustainability transitions in two different contexts. Beforehand, and based on the experience from the pre-test, we decided to choose two different country contexts that we would only choose during the session: the selection of the contexts was based on the two countries where most participants were coming from or living in, in order to create a sense of belonging to the respective context and to make the exercise eventually more meaningful for them. For this reason, Germany and Portugal were chosen as the country contexts.

**Table 1.** Research design.

| Topic | Hidden Knowledge about Dynamics in Sustainability Transitions Processes |
|---|---|
| Question/Goal | What are the different needs of selected social actors for a sustainable transformation in Germany and Portugal?<br>>Content-based: system visualizations and new assumptions concerning sustainability transitions and transformation<br>>Method-based: application of online constellation work to derive potentials for virtual collaboration and transdisciplinary sustainability research |

**Table 1.** *Cont.*

| Topic | Hidden Knowledge about Dynamics in Sustainability Transitions Processes |
|---|---|
| Constellation | Pre-test and online constellation during International Sustainability Transitions Conference 2020 (IST 2020, Dialogue session 546, http://ist2020.at/dialogue-sessions/ (accessed on 29 April 2021))<br>Issue-holders and session hosts: Antje Disterheft and Denis Pijetlovic<br>Facilitator: Antje Disterheft |
| Participants | Representatives: three experienced researchers/practitioners as participants of the conference IST 2020<br>Observers: 11 conference participants for the whole session<br>(14 participants in total, 2 session hosts) |
| Date | 21 August 2020 |
| Duration | 1.5 h (including an introduction and short reflection) |
| Research design | Exploratory research and action research |
| Data collection | Observation, screen shots, field notes, participants' feedback |
| Data preparation | Protocol, systematization of data, joint reflection |
| Data interpretation | Interpretative-hermeneutic method |

**Table 2.** Instructions given before the online constellation exercise for representatives and the audience, respectively.

| Instructions for Representatives | Instructions for Observers (Audience) |
|---|---|

(Note: Image was not visible for the representatives)

- You will be guided by the constellation facilitator. Please be open for the process.
- When invited to move, please take your time to test different positions (i.e., standing close to other elements or further away). Follow your intuition.
- You are invited to notice any body sensations you are feeling during the constellation process and to pay attention to any feelings, emotions, images, or metaphors that may arise.
- Please respect the structured dialogue: please share when asked to share and please do not interrupt others while speaking.
- Any movement is an intervention in the system and may influence the system and other elements. For the sake of clarity, please only undertake movements when invited to do so by the facilitator.
- Resist the curiosity to know which element you are representing and please do not look it up.

You are invited to observe the process with an open mind and heart.
Please take notes along the process, i.e.,:
What seems to be coherent to your (current) understanding or point of view?
What is strongly different from your expectations? What causes confusion or confounds you?
Coherence feels more pleasant and may comfort us, but confounds, i.e., anything that confuses or surprises us because it is different to what we expected, are interesting and may lead us to some hidden knowledge.

The elements were coded with colors: green for *science*, red for *politics*, and blue for *society* (the meaning of the codes was only visible to the audience to respect the double-blind format) (Table 2). The participants who volunteered to act as representatives were invited to choose a color and represent it by following their intuitive perception. The continuum as a representation for sustainability transitions was also coded with the number 1 for *weak sustainability narratives* and the number 2 for *strong sustainability narratives*, which were positioned opposite each other. The online constellation was led by the first author Antje

Disterheft. The process consisted mainly of letting the elements freely choose a position in the given contexts (by moving their icons in the prepared shared online slide) and eventually asking them if they can move from 1 (weak sustainability) towards 2 (strong sustainability), encouraging them to describe what they are sensing in the process. The exercise consisted of two phases. In the first phase, the elements described their perception for context A (Germany, only visible to the audience). In the second phase, there was a context change by moving to the next shared slide and the elements were then asked to describe their perception for context B (Portugal, also only visible to the audience). Only after the constellation had finished, the facilitator revealed to the representatives the elements they were representing and the two contexts. The session ended with a joint discussion and reflection with all participants on the exercise.

The data analysis and interpretation consisted of data systematization and joint reflections rounds between the authors, following an interpretative-hermeneutic approach [37].

## 4. Findings

In this section, we present the findings of the constellation exercise, namely how the representative elements acted in the respective context for Germany and Portugal. Since the main objective of this paper is to reflect about the online format and to extract potentials and lessons learnt of using exploratory constellations for scientific purposes, we do not intend to offer a deep content analysis of the constellations, but rather offer a traceable understanding of the process in order to then depict possibilities for new views and insights on sustainability transitions in the discussion section.

### 4.1. Context A—Germany

Figure 2 shows the final image of the first part of the constellation exercise with regard to sustainability transitions in the context for Germany. When asked to choose a position in this context that feels "right", all elements—*science, politics and society*—positioned themselves relatively immediately near the strong sustainability narrative; it was not necessary that the facilitator invited them to experiment with moving towards this point in a second step.

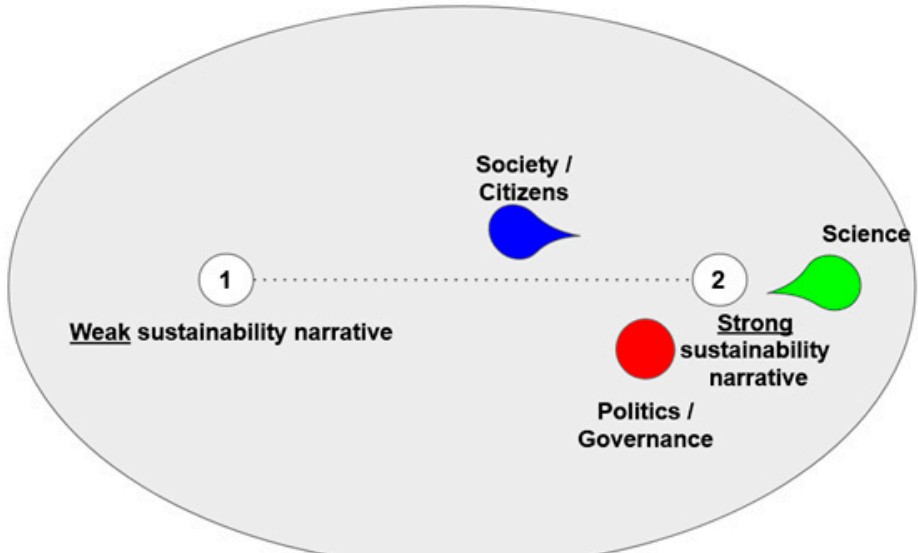

**Figure 2.** Sustainability transitions in Germany—final image of the representing elements' positions in the exploratory online constellation for the German context.

An interesting phenomenon that we could observe in this part of the online constellation was the self-perception and external perception of the element *science*. *Science* was the first and fastest element in choosing a position. The elements *society* and *politics* noticed

both this rapidity *of science*, whereas *science* itself did not notice this at all. In the context of Germany, *science* positioned itself behind the *strong sustainability narrative*. The other elements (*politics* and *society*) felt attracted by this positioning of *science* in the direction toward the *strong sustainability narrative*. Apparently, the element *science* triggered a pull effect (attraction) on the other elements. *Politics* described the phenomenon of the pull effect in the sense that the transgression of polarity by *science* has opened up a new space and this new space has a force of attraction, urging *politics* to also want to be a part of this structure. The element *politics* has made itself "round" for this; i.e., it changed its shape from an icon "with a nose" to a circle, and thus also the viewing direction of the icon had dissolved. The viewing direction is considered important in constellation work, as it can give information about the type of relationship one element has with another one (whether they are facing or looking at each other or not, or whether one stands with the back towards another element). In physical constellations with persons as representatives the viewing direction is easy to observe. In order to imitate the viewing direction, in online constellations icons" with a nose", i.e., some directing point, are chosen as these can help to understand better in which direction an element is looking or if elements turn away from other elements. The reason for dissolving its viewing direction was that *politics* in this context had the need to symbolize that it can hook up anywhere. In principle, it wanted to react flexibly, and the round shape could seemingly enable this need more effectively.

### 4.2. Context B—Portugal

Figure 3 shows the final image of the second part of the constellation exercise with regard to sustainability transitions in the context for Portugal. In this context, the role of element *science* was completely different. Here, *science* waited to see where the other two elements—*politics* and *society*—would position themselves in order to orient itself. *Politics* and *society* positioned themselves closely together and in proximity to the *weak sustainability narrative*. The element *politics* made its icon bigger than it actually was in the beginning, and the element *society* made its icon much smaller. *Science* then also felt the need to make itself bigger, even bigger as politics, in order to be noticed by the other two in this context. Furthermore, *science* placed itself rather behind the other two elements and wanted to push the other two elements more in the direction of a *strong sustainability narrative*. In the Portuguese context, *science* had a push effect. It expressed verbally the perception that the system only moves forward when it pushes *politics* and *society*.

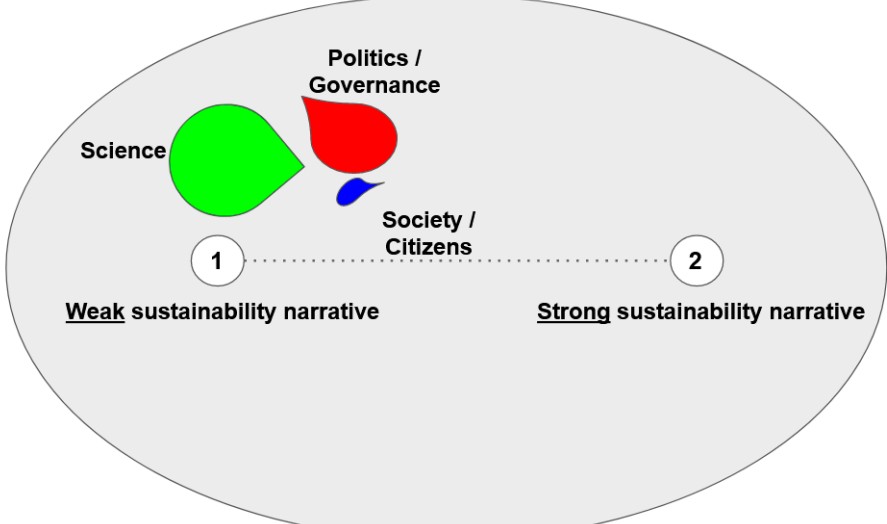

**Figure 3.** Sustainability transitions in Portugal—final image of the representing elements' positions in the exploratory online constellation for the Portuguese context.

All elements were invited to experiment briefly with moving closer to the *strong sustainability narrative*, but did not feel ready to do so. Due to time limits, such steps could not be explored further.

*4.3. Participants' Feedback*

The feedback we received in the final reflection round from the session's participants—both from the voluntary representatives and the observers of the constellation—were unanimously positive. Participants highlighted the double-blind format of the constellation (i.e., the representatives were unaware of the context and of what they represented) as the most surprising aspect. Further aspects discussed were the opportunities for a different communication, transferability into teaching and research (we shared further reading and material lists), and similarities to other methods such as Social Presencing Theater.

## 5. Discussion: On the Road of Discovery—Emerging New Assumptions and Questions

In this section, we aim to discuss the presented exercise from an explorative-guided perspective [38] and to invite the readers to a new type of seeing, allowing new reflections to emerge: what do we see that we did not see before? What is surprising or even confounding?

As explained in Section 2.1., exploratory constellations do not serve to picture reality, but offer visualizations of differentiations that can help to reflect on what was seen. There is a tendency, when it comes to interpretation and meaning-making, to first look for what makes sense to us. Confirmation bias, which refers to the attempt of seeking or interpreting data in a way that confirms or supports one's own existing beliefs [39,40], is a typical temptation in this endeavor. Exploratory constellations can be compared to being on a discovery trip with a group of people where each person takes their own pictures. Those pictures will have different angles and perspectives, and their diversity might be surprising, but they reflect how each photographer perceived the world. Luhmann [19,41] reminds us about our contingent worldviews and that everything could be different. With this in mind, we offer some lines of thought and new assumptions to trigger further discussion and reflection on the potential of exploratory constellations for sustainability transition research. Comparing Figures 2 and 3, we raise the assumption:

(i)　　A strong sustainability narrative does not (necessarily) lead to action and transformation.

We recall that, whereas in Germany, all represented elements positioned themselves already relatively close to a strong sustainability narrative—what already might be a surprise in itself—and science took a role to pull politics and society in coming even closer to this narrative, in the Portuguese context, all elements stood rather close to the weak sustainability narrative, and science seemed to push the other elements towards strong sustainability.

Germany apparently is able to afford a strong sustainability narrative. Funding might reflect this understanding, as German Research & Development (R&D) activities and related funding are highly above the OECD's and European Union's average [42]. Furthermore, Germany specifically promotes sustainability research with over 4 billion Euros, as it is stated in the strategy of the German Federal Ministry of Education and Research [43]. While being considered a "green pioneer" in many fields, and one of the few countries worldwide that has enshrined, e.g., climate neutrality by 2050 in its national law, Germany, however, struggles in keeping up with its sustainability goals and emission reduction targets [44]. Portugal, a world leader in integrating wind and solar energy [45], instead was one of nine EU member states that were on track towards their respective 2020 energy efficiency goals and stayed within its emission allocations [46]. It is not our intent to reduce the sustainability debate to country-specific energy performances and funding schemes, since historic and cultural differences also play their role. However, we might allow the question—eventually even contrary to our personal beliefs—whether the strong sustainability narrative is leading to the necessary action and transformation aspired for

in sustainability transitions. Why are more decisive actions being withheld? From the perspective of psychology, one could argue that knowledge alone does not automatically lead to action. This phenomenon is known as the *status quo bias* and describes a kind of inertia [47]. Despite knowledge (here: strong sustainability narrative), there is a strong psychological tendency to maintain the status quo or to prefer the given option above any change. If we take this approach further, then incentive systems, laws, and emotionality in the form of nudging [48] could be possibilities to counteract the status quo bias.

Furthermore, we might want to reflect on the types of relationships between science and society, science and politics, and politics and society, and include further actors in the picture. We can observe that in both contexts, science appears to strive for a leading role in sustainability transitions, but within different narratives. In the Portuguese context, science practically inflated itself whereas society made itself so small that it became less visible. Portuguese politics also increased its size and demanded more significance, but turned "its back" on the strong sustainability narrative, while German politics turned itself into a circle with no direction. What type of narrative might help society and politics, together with other actors such as, e.g., enterprises, to step up into bolder actions for structural change and transformation, overcoming, e.g., the status quo bias? These questions take us unto further reflections:

(ii)    Transformation requires integrating narratives beyond weak and strong sustainability

The findings presented in Figures 2 and 3 might suggest that the chosen narratives for this constellation setup are of less relevance for society, as the element in both contexts rather oriented itself on the positioning of the other elements than on identifying with a weak or strong sustainability narrative. Sustainability transitions research might want to put more attention on transformative narratives and create linkages to the leverage points for transformation (see Section 2.2), e.g., reconnecting people with nature [27].

Scholars speak about a "future narrative deficit" and a "narrative gap between our 'now' and visions for the future" [49]. Narratives are increasingly considered of great importance for sustainability, as they comprise a common socio-psychological infrastructure [50] and form "the basis for knowing how the world can be changed and manipulated (epistemology), while shaping the individual and cultural cognition that engenders a sense of being-in-the-world (ontology)" [49]. They are of particular importance in science, as they shape how we communicate [22] and provide unique opportunities to disrupt worldviews and universal rationality as "discourses on the unthinkable" [51]. The case of the Whanganui river, New Zealand, constitutes an example of such an unthinkable discourse, when after a 140-year-long litigation, the river was granted the same legal rights as human beings [52]. Can we imagine shifting narratives to personhood for non-human entities and see the transformative potential such narratives would offer [49,51]? Luederitz et al. [53] identified four archetypal narratives in sustainability transitions: (1) the green economy, (2) low-carbon transformation, (3) ecotopian solutions, and (4) transition movement; each with strengths and weaknesses. The authors then distinguish between deep and shallow intervention types of these narratives: ecotopian solutions and the transition movement with their focus on the intent and design of a system fall in the category of deep interventions, whereas the green economy and low-carbon transformation primarily consider the flows and parameters of a system and are therefore rather shallow interventions. Linking these categorizations to the constellation exercise, we could associate these deep interventions to a strong sustainability narrative and the shallow interventions to the weak sustainability narrative, even though Luederitz et al. [53] see transformative potential in each of the narrative archetypes. They point out that "the question is not which narrative is superior in accomplishing sustainability outcomes, but how sustainability mainstreaming can be facilitated across different narratives and intervention types" [53] (p.403). They suggest creating nested meta-narratives, as these provide "explicit learning opportunities for understanding how framings of sustainability transitions emerge and how they relate to seemingly contradictory pathways" [53] (p.403). We agree with their approach and argue that exploratory constellations can be extremely useful to help develop such

meta-narratives: First, exploratory constellations let systems "speak" by giving a voice to different actors or to even abstract ideas, as laid out in Section 2. Second, they can be set up over different spatial and time scales. Third, by generating images, metaphors, and new stories, exploratory constellations constitute a rich resource for meta-reflection and discussion, confronting us with our worldviews and holding the necessary space for ambiguity, paradoxes, contradictions, and, simultaneously, allowing something new to be seen.

Finally, by exploring the future, they can also contribute to narrowing the future narrative deficit. Since research with exploratory constellations invites us to observe and make differentiations, it shares similar ideas to the narrative research paradigm, namely "to build logics and moments for the emergence of difference as the basis of new understandings and practices" [50]. Such ideas would also link to second order transformation research and align with transcending current thinking and taking a multifaceted approach to change [28].

Limitations of This Study:

Our findings are based on data from a single systemic constellation session during a scientific conference with a restricted time-frame, thereby limiting the selection of representing elements for the constellation set up as well as the final debriefing with the session participants. Further elements, for example, representing the economic and financial system, as done in the pre-test, could provide additional input on the dynamics in the respective systems. The data generated are thereby small, and more sessions, in combination with a higher variety of participants beyond scientific communities, and more in-depth exchange is needed to consolidate the findings and to explore further the guiding questions for this research. We also acknowledge to have taken a rather epistemological approach in designing the constellation set-up (see Sections 2.2 and 3.1), in which we recognize our multiple roles as researchers and personal worldviews. We strived to compensate the shortcomings of this approach with a reflective attitude towards all statements made in this paper.

Our findings can also be regarded as of a pre-scientific nature, so-called insight-guiding theses [13], which can be transformed into hypotheses for further scientific investigation. In the theory of science [54], this process in research is referred to as "discovery contexts" and closely linked to creativity research [55]. Therefore, researchers may follow various procedures to develop new hypotheses. Systemic constellations can be one of the methods used in such procedures. They are a means to explore unresearched context in order to develop hypotheses that can then be examined further with other (traditional) scientific methods.

Furthermore, communicative post-validation of the findings would be an important step to make them more substantial in their value. Future exercises and experiments with online constellations could try to include more feedback from participants about theses, ideas, and assumptions of the research team, also encouraging the co-development of theses with participants in order to triangulate more effectively the results and findings.

## 6. Conclusions—Potentials of (Online) Exploratory Constellations for Sustainability Research

In this paper, we introduced exploratory systemic constellations as a research method that was tested during the online conference IST2020 in order to explore hidden dynamics related to sustainability transitions in Germany and Portugal.

The constellation exercises revealed interesting differences between the two contexts, particularly about the role of science in sustainability transitions, and served as an example for how the method can be used (virtually) as an invitation to see system dynamics from different perspectives and trigger new questions and assumptions.

Below, we summarize the potentials of (online) exploratory constellations for sustainability research:

(1). With regard to sustainability transitions and transformations, exploratory constellations constitute an excellent opportunity for interaction between academic and

non-academic actors and for exploring together the underlying dynamics for effective structural change. With the support of an experienced constellation facilitator, they can be fruitfully integrated into collaborative and transdisciplinary research;

(2). Constellations are excellent entry points for asking new questions;

(3). Exploratory constellations help develop meta-narratives and can therefore be powerfully combined with narrative research and second order transformation research, or similar approaches.

Potentials with regard to online format specificities:

(4). Online constellations allow a very flexible set-up that can relatively easy bring together not only people from diverse backgrounds, but also from different geographic locations, increasing the interaction between people who otherwise would be confronted with more difficulties or even barriers to meet. Practically, all the advantages of online collaboration apply, with the additional benefit of also exploring together the virtual space from a different perspective;

(5). Online constellations can visualize powerfully questions of significance. One novelty in the online format is that the representatives can not only move around, but change their shape and size, whereas in physical constellations, this would not be possible;

(6). Online constellations offer a constant bird-eye view of the system. Participants as representatives in a constellation exercise can position themselves in the field while continuously being able to see the whole system. Thereby, online constellations provide the unique opportunity for a second order observation, i.e., a self-reflective observation from inside and outside the system at the same time.

Exploratory systemic constellations can be a powerful method and tool on the road of discovery towards more sustainable futures.

**Author Contributions:** Conceptualization, A.D.; methodology, A.D., D.P.; validation, A.D., D.P., G.M.-C.; formal analysis, A.D., D.P., G.M.-C.; investigation, A.D., D.P.; data curation, D.P.; writing—original draft preparation, A.D.; writing—review and editing, A.D., D.P.; supervision, G.M.-C. All authors have read and agreed to the published version of the manuscript.

**Funding:** The first author received funding for this research by Fundação para a Ciência e Tecnologia (FCT), Portugal, grant number SFRH/BPD/115192/2016. The APC were funded by the University of Bremen, Germany.

**Institutional Review Board Statement:** Ethical review and approval were waived for this study, due to voluntary participation in the constellation exercise guaranteeing total anonymity and excluding the collection of any sensitive data. Participants were given the choice to opt-out at any moment.

**Informed Consent Statement:** Informed consent was obtained from all participants involved in the study.

**Acknowledgments:** We would like to thank IST 2020 and the participants in the dialogue session 546 for making this constellation exercises possible. We also would like to thank the participants from the pre-test group for their availability and valuable input in the discussion rounds. Furthermore, we are grateful to the University of Bremen for covering the APC fees. The first author acknowledges and thanks the support given to CENSE by the Portuguese Foundation for Science and Technology (FCT) through the strategic project UIDB/04085/2020. Finally, we thank the three anonymous reviewers for their constructive comments on an earlier version of this paper.

**Conflicts of Interest:** The authors declare no conflict of interest.

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
