# Peer review of "On the Road of Discovery with Systemic Exploratory Constellations: Potentials of Online Constellation Exercises about Sustainability Transitions"

_sustainability, doi:10.3390/su13095101_

Round 1

Reviewer 1 Report

  • this is a very interesting research
  • please shorten chapter 2.2
  • chapter 2.3 should be deepened in terms of your original research, e.g. please introduce the different elements (politics, science, society) and why they are important in your research frame
  • deepen transformation science and reduce constellation work - set a clear content-based frame and focus
  • link scientific facts and theory more strictly with your constellation outcome - reflect it more precisely and embed your limitations in your discussion
  • Please justify why you did not integrate companies as they have to follow weak or strong sustainability, in line finance or the financial system - is the chosen system not rather limited and restricted in giving answers to your research questions?
  • Why did you distinguish Germany and Portugal? Please justify your selection criteria
  • chapter 5 - limitations - please widen your limitations, incl. referring to the scientific quality criteria
  • the paper should be proofread by an English native

Reviewer 2 Report

It is a an interesting idea to use exploratory systemic constellations as a support tool to manage transitions. However, because  participants in the experience are mainly people from academia and a much more diversified set of actors were excluded as a result of the pre-test, the exercise suffers from major limitations that should be discussed. In particular how having academia representing 3 actors affects the results of the exercise? 

We also feel that more detail on how the experiment was organised should be given! How many people participated in each context? How many participants in the exercise were non-academics?

The labeling also needs further explaining ? e.g. Because governance is a structure and a process "Politics/Governance" is a confusing label for an actor.  

In addition, it looks like we would not need the exercise at all to understand the limited relevance of the dichotomy between weak and strong narratives as a basis for a useful "transformative narratives" ?

Reviewer 3 Report

I'm very happy to have the opportunity to review this paper. I've read your paper as an attempt of showing systemic constellations as a method for researching sustainability transitions. As such I have  I would like to share some of my comments:

  • I would expect more detailed explaination of what is weak and strong sustainability [perception?] It is important for proper understanding of your research design.  I would also like to know, how did you assured the common understanding of the categories among participants: weak and strong sustainabiity narrative? How is this important for systemic constellations?
  • While explaining the methodology I would expect that you explain in details the procedure while chosing the experiment context, neccesary conditons, etc. You provide general information on the topic of the Conference that set the context for this research, however I think it would be crucial to explain the procedure for idenfitying the specific context for sustainability transitions research. Questions as: what kind of conference was this and how it is linked to the research, how tho chose the specific context for such research, etc. should be answered.  It seems important to explain how the topic of your research relates to the research question and why did you chose this particular conference. You could provide also specific data on how to generate / gather the relevant data. From your description I understand that the research was limited to the 1,5 hour. Was this enough to gather the relevant data? Could you explain in details how this "short feedbeck and reflection round" relates to the quality of data you could gather?  How do you assured the accuracy of data gathered? I understand that you had no influence on the acceptance of the conference members, does it play a role in conducting systemic constellations?  
  • In my opinion the process of sampling seems important fot the experiment success, but you provide little or no background on the German and Portugal sustainable transitions (first in discussion section). I understand the practical reason for chosing this countries ("participants came from this regions"), however I would like to know what were the main assumptions regarding the differences in sustainability constellations. 
  • You specify  in the beginning of section 4 that this paper is methodological one therefore you don't provide specific information on content analysis. However you also don't provide any specific information on how to conduct the experiment properly. You describe how you did it, but I right now there is for me too many open question left, so that I'm not convinced how this method can deliver quality data. 
  •  Of course from the perspective of the scope of the "Sustainability" journal I'm very  interested in the findings from this experiment (these seem as interesting topic for a different article). However if you position your paper as educational/methodological research I would recomment to add much more rigours methodological background and justifications, eg. why Science, Politics and Society were chosen, how to select the right participants, how to assure the proper understanding of the task, etc. In my opinion it requires therefore rewriting the section on findings (partially), discussion and limitation as you skipped there to presenting proper discussion of results (which you have not presented and did not intended to present). 
  • It is crucial to justify how this kind of experiments may be conducted in order to gather qualitative data of high quatliy for research purposes, discuss which research purposes could be apropriate for this kind of research design, etc. In the paper you rather took an descriptive approach so partially you miss the link to the goal of the paper.
  • To sum up I would expect more deeper explaination and critical presentation of the method and its limitation, providing methodological guidelines and accuracy of this methodology for explaining the sustainability research.

Good luck!

Round 2

Reviewer 3 Report

Thank you very much for all the clarification. In the current verstion the paper, tha method and its novelty is for me much more clear.